# Real-Time Fluorescence Image-Guided Oncolytic Virotherapy for Precise Cancer Treatment

**DOI:** 10.3390/ijms22020879

**Published:** 2021-01-17

**Authors:** Shuya Yano, Hiroshi Tazawa, Hiroyuki Kishimoto, Shunsuke Kagawa, Toshiyoshi Fujiwara, Robert M. Hoffman

**Affiliations:** 1Department of Gastroenterological Surgery, Okayama University Graduate School of Medicine, Dentistry and Pharmaceutical Sciences, Okayama 700-8558, Japan; htazawa@md.okayama-u.ac.jp (H.T.); kishipon@ga2.so-net.ne.jp (H.K.); skagawa@md.okayama-u.ac.jp (S.K.); toshi_f@md.okayama-u.ac.jp (T.F.); 2Center for Graduate Medical Education, Okayama University Hospital, Okayama 700-8558, Japan; 3Center of Innovative Clinical Medicine, Okayama University Hospital, Okayama 700-8558, Japan; 4Minimally Invasive Therapy Center, Okayama University Hospital, Okayama 700-8558, Japan; 5AntiCancer, Inc., San Diego, CA 92111, USA; all@anticancer.com; 6Department of Surgery, University of California, San Diego, CA 92093, USA

**Keywords:** cancer cell cycle, fluorescent proteins, FUCCI, imaging, adenovirus, oncolytic virotherapy, cancer stem cell, mouse model, orthotopic

## Abstract

Oncolytic virotherapy is one of the most promising, emerging cancer therapeutics. We generated three types of telomerase-specific replication-competent oncolytic adenovirus: OBP-301; a green fluorescent protein (GFP)-expressing adenovirus, OBP-401; and Killer-Red-armed OBP-301. These oncolytic adenoviruses are driven by the human telomerase reverse transcriptase (hTERT) promoter; therefore, they conditionally replicate preferentially in cancer cells. Fluorescence imaging enables visualization of invasion and metastasis in vivo at the subcellular level; including molecular dynamics of cancer cells, resulting in greater precision therapy. In the present review, we focused on fluorescence imaging applications to develop precision targeting for oncolytic virotherapy. Cell-cycle imaging with the fluorescence ubiquitination cell cycle indicator (FUCCI) demonstrated that combination therapy of an oncolytic adenovirus and a cytotoxic agent could precisely target quiescent, chemoresistant cancer stem cells (CSCs) based on decoying the cancer cells to cycle to S-phase by viral treatment, thereby rendering them chemosensitive. Non-invasive fluorescence imaging demonstrated that complete tumor resection with a precise margin, preservation of function, and prevention of distant metastasis, was achieved with fluorescence-guided surgery (FGS) with a GFP-reporter adenovirus. A combination of fluorescence imaging and laser ablation using a KillerRed-protein reporter adenovirus resulted in effective photodynamic cancer therapy (PDT). Thus, imaging technology and the designer oncolytic adenoviruses may have clinical potential for precise cancer targeting by indicating the optimal time for administering therapeutic agents; accurate surgical guidance for complete resection of tumors; and precise targeted cancer-specific photosensitization.

## 1. Introduction: Intravital Imaging

High-resolution computed tomography (CT), magnetic resonance imaging (MRI), positron emission tomography/CT (PET/CT), and ultrasonography (US) are indispensable to detect tumors and evaluate treatment efficacy in the clinic. Powerful non-invasive fluorescence imaging technologies, including confocal laser-scanning microscopy, and multiphoton laser-scanning microscopy have been developed to visualize cancer at the subcellular level in mouse models. Shimomura discovered green fluorescent protein (GFP) from the jellyfish *Aequorea* which can brightly label cells and proteins [1]. Tsein et al. developed bright GFP mutants and GFP mutants that acquired new colors, including cyan, blue, orange, and yellow fluorescence, which enabled multispectral in vivo as well as in vitro imaging [2,3]. Fluorescent-protein genes were genetically linked to a targeted gene whose function could then be visualized in vivo as well as in vitro [4,5]. Multicolor fluorescent mutants and sophisticated imaging instrumentation and software enabled researchers to detect the location of cancer cells and monitor their molecular functions and dynamics in mouse models [6,7,8,9,10,11,12,13].

Oncolytic virotherapy is a promising, emerging targeted therapy [14,15,16,17,18,19,20]. We developed three types of cancer-targeting oncolytic adenoviruses that are driven by the hTERT promoter: OBP-301, OBP-401, and OBP-301 with KillerRed (Figure 1). The hTERT promoter, highly expressed in most cancers, drives the adenoviral E1A and E1B genes, which are essential for virus replication. OBP-301 is effective against most solid tumors that have high telomerase activity (Figure 1A) [21,22,23,24,25]. OBP-401 contains the *GFP* gene, enabling preferential labeling of cancer cells (Figure 1B) [26]. OBP-301 containing KillerRed has been used for photodynamic therapy (PDT) of cancer cells (Figure 1C) [27]. In the present review, we focus on fluorescence-guided oncolytic virotherapy that can decoy quiescent chemoresistant cancer stem cells to cycle and become chemosensitive; fluorescence-guided surgery (FGS) with adenovirus-mediated GFP; and cancer-targeted PDT with adenoviral KillerRed.

## 2. Fluorescence Ubiquitination Cell Cycle Indicator (FUCCI) Imaging Guides Precise Timing of Cancer Chemotherapy after Adenoviral S-Phase Cell-Cycle Decoy

### 2.1. FUCCI Imaging Visualizes Quiescent Cancer Stem Cells (CSCs) Resistant to Conventional Therapy

Most cytotoxic chemotherapy targets cycling cancer cells [28]. However, conventional cytotoxic therapeutics have limited efficacy for solid cancers. Recent studies showed that cancer stem cells (CSCs) in solid cancers, as well as leukemia, play important roles in malignant progression, resistance to chemotherapy and radiotherapy, and recurrence [29,30,31]. CSCs are mostly quiescent in the G_0_/G_1_ phase; therefore, they are resistant to most currently-used cytotoxic chemotherapy and radiotherapy since they target proliferating cancer cells in S-, G_2_-, and M-phases [32,33,34,35]. Sakaue-Sawano et al. developed the FUCCI system for visualizing cell-cycle dynamics of individual cells in real time [36]. The FUCCI system uses genes linked to different-color fluorescent reporters, such as Cdt1 and Geminin, which only appear in specific phases of the cell cycle. The original FUCCI system comprised two plasmids: Cdt1-mKO2 (red fluorescent protein), expressed as red fluorescence in G_1_-phase, and Geminin linked to mAG (green fluorescent protein), expressed as green fluorescence in late S-phase. In early S-phase, when the expression of Cdt1-mKO2 is at the same level of Geminin-mAG, the cells appear yellow. Therefore, FUCCI-red indicates the quiescent G_1_ phase, FUCCI-green indicates the proliferating late-S/G_2_/M phase, and FUCCI-yellow indicates the early S phase [36]. The indicators in the next generation FUCCI series have been modified to distinguish all phases of the cell cycle [37,38,39].

FUCCI imaging demonstrated that CSCs remain quiescent in G_0_/G_1_ phase for longer periods than non-CSCs [40]. For example, FUCCI imaging showed that CSC-derived tumor spheres which comprise quiescent cancer cells expressing FUCCI-red, can remain quiescent for more than 14 days, and are resistant to conventional chemotherapy and radiotherapy [40] (Figure 2A).

### 2.2. FUCCI Imaging Demonstrates that an Oncolytic Adenovirus Decoys Quiescent Cancer Stem Cells to Cycle to S-Phase. Cytotoxic Chemotherapy Can then Be Presicely Administrated to Eradicate the Decoyed Cancer Cells

Adenoviruses are known to decoy infected cells to cycle and be trapped in S-phase during viral DNA synthesis. The viral E1 gene can induce cellular cyclin E and other cell-cycle genes that decoy the infected cells to cycle from G_0_/G_1_ to S-phase [41,42,43]. We demonstrated with real-time FUCCI imaging that oncolytic adenovirus OBP-301 decoyed quiescent CSC-derived tumor spheres to cycle in vitro [40]. FUCCI in vitro imaging demonstrated that decoyed cancer stem cells became sensitive to conventional chemotherapy [40]. FUCCI in vivo imaging demonstrated that small and nascent tumors comprise mostly proliferating cancer cells; in contrast, quiescent cancer cells comprise the vast majority of cells in established tumors. In established tumors, proliferating cancer cells are located only on the surface of the tumor or adjacent to tumor blood vessels [40]. FUCCI in vivo tumor imaging also demonstrated that conventional cytotoxic chemotherapy killed only proliferating cancer cells, and quiescent cancer cells were resistant to currently-used cytotoxic chemotherapy and restarted proliferation after cessation of chemotherapy [44,45]. This is one of the reasons why currently-used chemotherapy has limited efficacy against solid cancers [46,47]. OBP-301 was shown to decoy quiescent cancer cells within tumors to cycle with the decoyed tumors becoming sensitive to chemotherapy [40]. The above results showed that an oncolytic adenovirus can decoy quiescent cancer stem and non-stem cells, sensitize them to conventional chemotherapy, whereby they are killed by oncolysis and chemotherapy. FUCCI cell-cycle imaging guides the precise timing for chemotherapy to be administered to effectively eradicate the decoyed cancer cells [46,47] (Figure 2B).

## 3. Oncolytic GFP Reporter Virus for Fluorescence-Guided Surgery (FGS)

### 3.1. Fluorescence-Guided Surgery (FGS) with Adenoviral GFP: Efficacy and Limitations

Surgical resection is still the most effective therapy for most solid cancers. However, even radical resection of malignant tumors often leads to recurrence due to difficulties to correctly visualize the tumor margin [48,49]. Therefore, real-time intraoperative fluorescence guidance is ideal for precise and complete surgical resection of malignant tumors with a clear margin. FGS is an emerging technology [50,51,52,53,54,55,56,57,58,59]. Indocyanine green (ICG) and methylene blue, both of which were approved by the U.S. Food and Drug Administration (FDA), are used clinically in some large medical centers [60,61]. ICG was first used to detect sentinel lymph nodes (SLNs), which are the first draining lymph nodes, of breast cancer and malignant melanoma [62,63,64]. Positive SLNs indicate that tumors are metastatic, since breast cancer and malignant melanoma with metastasis to SLNs often metastasize to distant organs: liver, lung, brain, or bone [62,63,64]. Near-infrared (NIR) FGS is being used clinically in some medical centers as it penetrates more deeply in tissues than fluorescence in the visible light range. Anti-carcinoma embryonic antigen (CEA) and cancer antigen (CA)19-9 antibodies, and bevacizumab, conjugated with NIR fluorophores, have been used preclinically and are beginning to be used clinically for gastrointestinal surgery [65,66,67,68]. In a few large medical centers, 5-aminolevulinic acid (5-ALA) is used for fluorescence-guided surgery of highly invasive glioblastoma multiforme (GBM). Treatment with the metabolite 5-ALA, a precursor of hemoglobin, resulted in the accumulation of porphyrins within malignant glioma. Patients who underwent FGS with 5-ALA had longer 6-month progression-free survival rates (41%) than those who had bright light surgery (BLS) (21%) [69,70,71,72]. However, FGS with these agents has limitations because ICG, methylene blue, and 5-ALA are not specific to malignant tumors. Although fluorescent-antibody-based FGS is now emerging clinically, cancers, such as those of stomach, do not yet have specific antibodies for FGS.

We developed the GFP-expressing oncolytic adenovirus OBP-401, which preferentially labels cancer cells in vivo as well as in vitro [26]. OBP-401 has been used to detect sentinel lymph nodes in orthotopic colon-cancer mouse models [26]. OBP-401 has been used to detect peritoneally-disseminated cancer cells in mouse models, and subsequently, OBP-401-based FGS enabled resection of more cancer cells than bright-light surgery (BLS) [73]. Although OBP-401-based-FGS is effective on human tumors in mouse models, it has limitations for future clinical utility since unlike mouse models, adenoviruses can replicate in normal human tissue.

### 3.2. OBP-401-Based-FGS for GBM with Oncolysis of Invading GBM Cells

OBP-401 labeled invading glioma cells with GFP and killed them in two-dimensional culture and three-dimensional culture [74]. OBP-401-based FGS was used for GBM in an orthotopic mouse model with a red fluorescent protein (RFP)-expressing glioma cell line, U87GM-RFP. The orthotopic GBM tumor invaded the brain tissue rendering the margin between the tumor and normal brain tissues unclear under bright light [74]. BLS could therefore not completely resect GBM tumors. Intratumoral injection of low-dose OBP-401 (1 × 10^8^ PFU) labeled GBM with GFP and clearly highlighted the margin between the glioma and normal brain tissue. OBP-401-based FGS therefore enabled resection of the GBM tumor. High-dose administration of OBP-401 (2 × 10^8^ PFU) significantly reduced the size of the GBM tumor compared with low-dose OBP-401 (high-dose vs. low-dose: *p* < 0.05) [74]. The GBM was more effectively resected with high-dose OBP-401-based FGS than with low-dose OBP-401-based FGS or BLS. Twelve of 14 mice that underwent BLS of their GBM had tumor recurrence, compared to 5 of 14 that underwent low-dose OBP-401-based FGS (*p* = 0.002). High-dose OBP-401-based FGS of GBM resulted in no recurrence for more than for 120 days. Curative FGS of invasive glioblastoma (GBM) with preservation of brain function in the orthotopic mouse model was enabled by OBP-401 [74]. However, as mentioned above, adenoviral FGS has limitation in the clinic due to adenoviral replication in normal human tissues (Figure 3A).

### 3.3. Function-Preserving FGS with OBP-401 for Soft Tissue Sarcoma

Precise resection of soft tissue sarcoma (STS) is still one of the most difficult surgeries since the border between the STS and normal tissue is unclear. Therefore, there is frequent local recurrence of STS [75,76,77,78,79]. According to clinical recommendations, a surgical margin should be more than 5 cm from the tumor margin, which is often impossible [80,81]. Although FGS is beneficial as a treatment for STS, there are not any specific markers for tagging STS. OBP-401 brightly illuminated and then effectively killed HT1080 fibrosarcoma cells in vitro and in vivo [82]. We then demonstrated that OBP-401-based FGS improved surgical resection and disease-free survival using an orthotopic mouse sarcoma mouse model [82]. We developed the orthotopic model of STS by injecting HT1080 cells into muscle tissue since HT1080 cells have the potential to grow rapidly, invade the surrounding muscle, and metastasize to the lung(s). Conventional BLS could not completely resect the STS in this model [82]. Low-dose OBP-401 FGS improved STS over BLS also visualized residual sarcoma cells remaining after FGS. High-dose OBP-401 resulted in complete resection of STS without residual sarcoma cells in the surgical bed and with muscle-function preservation [82]. OBP-401-based FGS also inhibited lung metastasis compared with BLS in the orthotopic HT1080 model. All mice which received high-dose OBP-401 FGS survived for more than 120 days. In contrast, 50% of the mice with BLS were dead at less than 50 days. OBP-401-based FGS significantly improved disease-free survival and overall survival [82]. Thus, OBP-401-mediated FGS with the dual function of cancer-specific bright illumination and oncolysis is promising, but it needs to be demonstrated in future clinical studies that OBP-401 does not label normal tissues (Figure 3B).

### 3.4. FGS with OBP-401 for Osteosarcoma without Specific Markers

The standard treatment of osteosarcoma comprises wide resection with or without adjuvant radiotherapy and/or chemotherapy [83,84,85]. The residual cancer cells in the surgical area determine local recurrence, future distant metastasis, and prognosis [85]. However, there is still no specific marker for osteosarcoma for fluorescent-antibody-based FGS. OBP-401 illuminated RFP-expressing 143B (143B-RFP) and HNNG/HOS (HNNG/HOS-RFP) osteosarcoma cells with GFP in a dose-dependent manner [86]. We developed an orthotopic osteosarcoma model by implantation of 143B-RFP cells into the intramedular cavity of the left tibia of athymic nude mice [87]. The orthotopically-growing osteosarcoma cells extensively grew in the tibial bone. OBP-401 conferred GFP fluorescence to the orthotopic osteosarcoma tumor; which was sufficiently bright to perform complete resection with FGS [86]. After OBP-401-based FGS, there were no residual cancer cells. Moreover, OBP-401-based FGS reduced lung metastasis after surgery [86]. OBP-401-based FGS controlled the local recurrence and distant metastasis and therefore prolonged survival in this model. [86].

### 3.5. OBP-401-Based FGS for Liver and Lung Metastasis

Liver tumors such as hepatocellular carcinoma and liver metastasis, are detectable by infrared light after intravenous injection of ICG in the clinic. ICG-fluorescence imaging can be used to readily identify liver tumors in real time for open and laparoscopic surgery in the clinic [88,89,90,91]. However, ICG is not tumor specific.

We applied OBP-401 to FGS for metastatic liver tumors and for primary and metastatic lung tumors [92,93] (Figure 4). We first developed an orthotopic solitary metastatic model with a colon cancer cell line by implantation of HCT-116-RFP cells under the serosa of the liver of nude mice [92]. The orthotopic tumor grew extensively in the liver. Intratumoral and intravenous injection of OBP-401 visualized the metastatic liver tumor [92,94]. OBP-401-derived GFP and cancer-cell-derived RFP fluorescence precisely colocalized [92]. OBP-401-GFP enabled illumination and detection of cancer cells at the single-cell level. OBP-401-based FGS resulted in no detectable residual cancer cells in the surgical bed [92].

Goldstraw et al. reported no difference in mortality and local recurrence between open resections and video-assisted thoracoscopic surgery (VATS) [95,96]. VATS for lung cancer is now the standard procedure for non-small cell lung cancer. However, a currently-used marker for small lung tumor with a needle and a suture still provides only the rough location of a targeted tumor in the lung [97,98,99,100,101,102,103,104]. It is therefore difficult to know the precise location of a targeted tumor. Thus, the VATS marker must be improved. We developed an orthotopic solitary lung tumor model by direct implantation of an RFP-expressing tumor on the lung [93]. We detected the RFP-expressing tumor on the lung over the thorax one week after implantation. OBP-401 was injected into the tumor transthoracically by RFP guidance [93]. We detected OBP-401 illuminating the tumor 3 days after transthoracic injection of OBP-401. The chest wall of mice was opened with assisted ventilation, then we readily detected fluorescent A549 and H460 lung tumors and identified the precise location of the tumor and margin between the cancer and normal lung tissue. OBP-401-based FGS was performed to completely resect the lung cancer [93] (Figure 4). To further validate OBP-401 FGS on other tumor types, we performed OBP-401-based FGS on aggressive triple-negative breast cancer [105,106] and malignant melanoma [107].

## 4. Preclinical FGS with Patient-Derived Orthotopic Xenograft (PDOX) Mouse Models

We have demonstrated that OBP-401 is useful for FGS described above. However, the mouse models described above are based on cancer cell lines, not clinical samples. Patient-derived xenograft (PDX) models are currently being widely used for molecular biological studies, drug evaluation, and development of new strategies since PDXs maintain histologic, genetic, and epigenetic characteristics of their donor tumors in live mice [108]. We have developed patient-derived orthotopic xenograft (PDOX) models of colon cancer, pancreatic cancer, lung cancer, many kinds of sarcoma, and other tumor types in the past 30 years. PDOX models mimic cancer behavior in the human body better than the subcutaneous PDX models [109,110,111,112,113]. OBP-401 brightly labeled pancreatic cancer PDX and PDOX mouse model with GFP as clearly as in cell-line-derived xenografts [114]. OBP-401 did not label normal tissues in the PDX and PDOX models. The cancer-specific OBP-401-GFP fluorescence enabled FGS in a PDOX model of pancreatic cancer. The pancreatic tumor margin visualized by OBP-401 was much clearer than that under bright light [114]. FGS in the PDOX mouse model suggested that OBP-401 FGS is ready for trial in the clinical setting.

## 5. Photodynamic Virotherapy with OBP-301 Expressing KillerRed

Photodynamic therapy (PDT) is a nontoxic photosensitizer with harmless visible light to produce reactive oxygen species (ROS) in the cancer and subsequently causes cytotoxicity in cancer cells [115,116]. Photofrin, talaporfin, and 5-ALA are currently used for PDT against various cancers in clinical settings [116,117]. However, these photosensitizers have an unsolved problem: the lack of sufficient delivery to tumors, thereby limiting efficacy [116]. A red fluorescent protein, KillerRed, generates ROS under irradiation from a green fluorescent light or green laser [117,118]. The KillerRed protein is one of the brightest red fluorescent proteins and penetrates more deeply than GFP [118,119]. We generated OBP-301 containing KillerRed [27]. Compared with conventional photosensitizers, OBP-301 containing KillerRed has high cancer specificity and prolonged tumor accumulation of KillerRed since the virus replicates and produces a large amount of KillerRed protein only in infected cancer cells [27]. OBP-301 containing KillerRed maintains the virus-mediated cancer-cell-death efficacy as OBP-301. OBP-301 containing KillerRed along with light irradiation had greater efficacy against lymph node metastasis of colon cancer and orthotopic melanoma in mouse xenografts [120]. More pre-clinical studies need to be carried out with KillerRed to determine its general and optimal efficacy.

## 6. Conclusions

Viral therapy with a telomerase-dependent adenovirus is a promising cancer therapeutic strategy. Highlights of telomerase-dependent viral therapy include decoy of quiescent cancer cells to cycle which makes them chemosensitive; FGS and PDT. Phase I clinical trials in the United States showed that OBP-301 was safe for patients with advanced solid tumors [22]. A Phase I clinical study of combination therapy with OBP-301 and radiotherapy has been conducted to confirm safety, tolerability, and feasibility on patients with esophageal cancer in Japan [25]. Future clinical studies will be performed for anti-tumor efficacy with telomerase-dependent adenovirus.

## Figures and Tables

**Figure 1 ijms-22-00879-f001:**
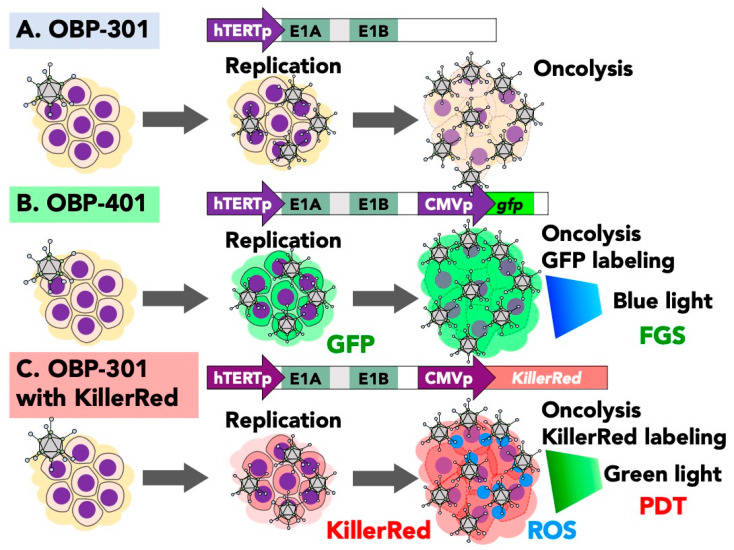
Telomerase-specific conditionally-replicating oncolytic adenoviruses. (**A**) OBP-301 is driven by the hTERT promoter for preferential replication in cancer cells. OBP-301 replicates preferentially in cancer cells that have high expression of telomerase. OBP-301 induces cancer-specific cell death after replication. (**B**) OBP-401 contains a cassette comprising of a CMV promoter and green fluorescent protein (*GFP*) gene. OBP-401 preferentially labels cancer cells with GFP and then kills them, depending on the multiphcity of infection. (**C**) OBP-301 with KillerRed contains a cassette comprising the CMV promoter and KillerRed gene. This virus preferentially labels cancer cells with KillerRed. OBP-301 with KillerRed has a dual function of adenovirus-mediated cancer-cell killing and PDT.

**Figure 2 ijms-22-00879-f002:**
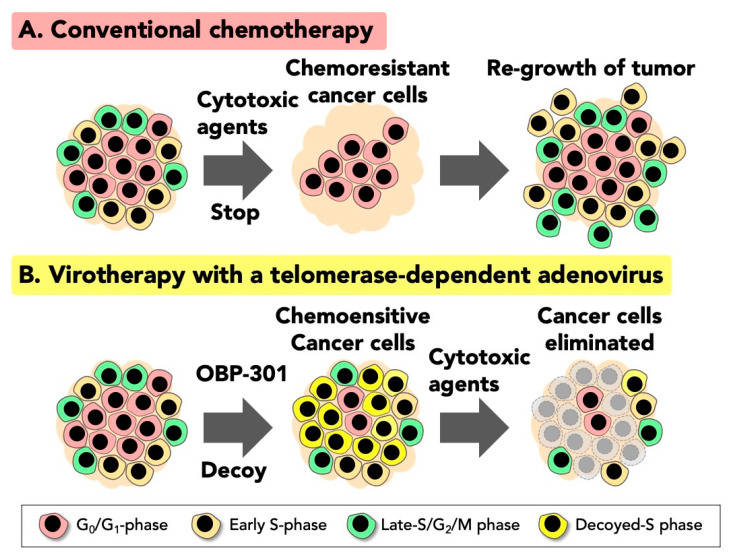
Fluorescence ubiquitination cell cycle indicator (FUCCI) imaging demonstrates that an oncolytic adenovirus decoys quiescent cancer stem cells to cycle and become sensitive to chemotherapy. (**A**) Conventional chemotherapy only kills proliferating cancer cells in the S/G_2_/M phase. In contrast, chemotherapy has little efficacy against quiescent cancer cells in G_0_/G_1_ phase. Resistant quiescent cancer cells restart proliferation after chemotherapy. (**B**) Oncolytic adenovirus, OBP-301, decoys quiescent cancer cells to cycle to early S-phase which makes them sensitive to chemotherapy. Conventional cytotoxic chemotherapy has great efficacy against decoyed cancer cells in S-phase.

**Figure 3 ijms-22-00879-f003:**
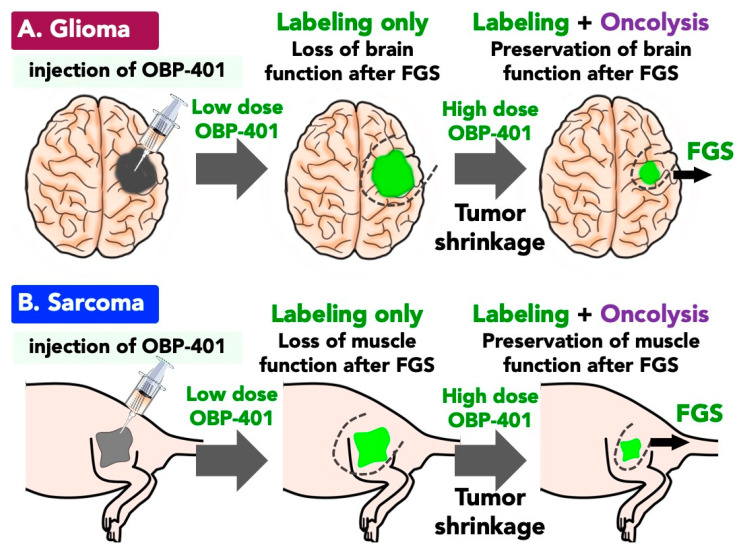
OBP-401-GFP mediated fluorescence-guided surgery (FGS) with preservation of function. (**A**) FGS for glioblastoma (GBM) with OBP-401-mediated bright GFP tumor labeling and oncolysis. Whole-tumor imaging with an orthotopic GBM mouse model demonstrated that low-dose OBP-401 labels GBM tumors with GFP and clearly highlights the margin between the GBM and normal brain tissue, which enables FGS with partial loss of brain function. High-dose OBP-401 shrinks the GBM tumor, then enables effective FGS with preservation of function. (**B**) FGS for soft tissue sarcoma (STS) with OBP-401-mediated bright GFP tumor labeling and oncolysis. Whole-tumor imaging with an orthotopic STS mouse model demonstrated that low-dose OBP-401 labels STS tumors with GFP, clearly highlights the margin between the STS tumor and normal muscle or fat tissue, and consequently enables FGS partly with partial loss of motor function. High-dose OBP-401 shrinks the STS tumor, then enables effective FGS with preservation of motor function.

**Figure 4 ijms-22-00879-f004:**
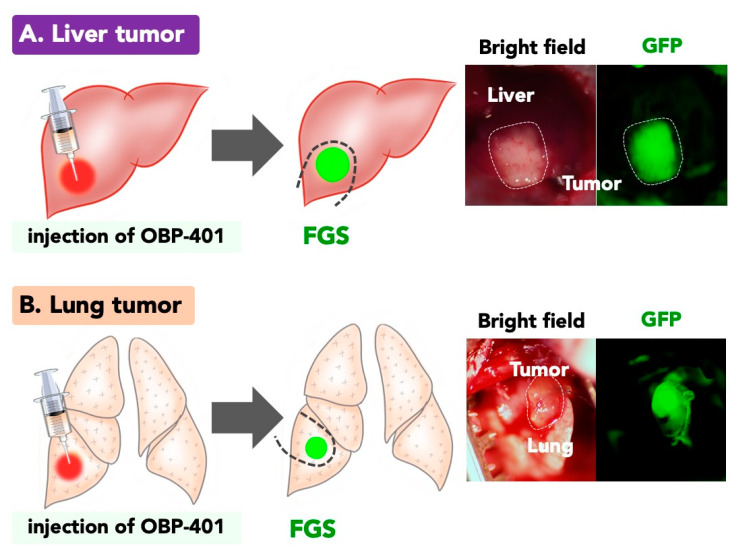
OBP-401-mediated GFP-guided FGS for liver tumors and lung tumors. (**A**) OBP-401 is injected in the liver tumor under RFP guidance before surgery. OBP-401 labeled the liver tumor with GFP and clearly highlighted the margin between the tumor and normal tissue, enabling complete resection. Representative image of a liver tumor labeled with OBP-401 in orthotopic mouse model. (**B**) OBP-401-FGS for lung metastasis. OBP-401 was injected in the lung tumor transthoracically under RFP-guidance before surgery. OBP-401 labeled the lung tumor with GFP and clearly highlighted the margin between the lung tumor and normal tissue. Representative image of a lung tumor labeled with OBP-401 in an orthotopic mouse model.

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
