# Peer review of "Real-Time Fluorescence Image-Guided Oncolytic Virotherapy for Precise Cancer Treatment"

_ijms, 2021, doi:10.3390/ijms22020879_

Round 1

Reviewer 1 Report

The review recapitulates the findings of Yano et al. The group has previously generated three adenovirus constructs that replicate in cancer cells with an active hTERT. The addition of fluorescent dyes expanded the use of oncolytic virus constructs from killing the cancer cells to assist in tumour excision by precisely delimiting the border between cancer cells and normal tissue.

The review is well structured and the findings are relevant to the oncolytic virotherapy field.

However, the manuscripts should be extensively edited for English language and style.

Other comments:

Line 35: intravital and in vivo mean the same thing, should use either one of the formulations

Line 73: FUCCI should be explained as it is the first time the abbreviation shows up in the text

Lines 87-89: FUCCI orange is later referred as FUCCI red

Line 139: repetition of NIR fluorescence

Line 185: GMB mice model should be mouse model

Line 209: significantly improved disease-free survival, should mention how long the survival increased

Lines 225-231: should be more succinct

Line 239: would be helpful to shortly mention the findings of the studies cited

Line 244: intratumorally and intravenously should be intratumoral and intravenous injection 

Line 251: VATS maker should be marker 

Lines 275-288: the paragraph is about PDX but does not explain what host species has been used until the last sentence. 

Line 288: mice should be mouse

This is not an exhaustive list of corrections. Please review the manuscript carefully for repetitions, redundancies, explanations of abbreviations, consistency, etc.

Author Response

Reviewer #1

Summary

The review recapitulates the findings of Yano et al. The group has previously generated three adenovirus constructs that replicate in cancer cells with an active hTERT. The addition of fluorescent dyes expanded the use of oncolytic virus constructs from killing the cancer cells to assist in tumour excision by precisely delimiting the border between cancer cells and normal tissue.

The review is well structured and the findings are relevant to the oncolytic virotherapy field.

However, the manuscripts should be extensively edited for English language and style.

Comment 1;

Line 35: intravital and in vivo mean the same thing, should use either one of the formulations

Response:

Line 36: We deleted the word “in vivo” in the revised version.

Comment 2;

Line 73: FUCCI should be explained as it is the first time the abbreviation shows up in the text

Response:

Lines 24-25: We have defined the abbreviation FUCCI in the revised version.

Comment 3;

Lines 87-89: FUCCI orange is later referred as FUCCI red

Response:

Dr. Sakaue-Sawano uses red fluorescent protein named as Kusabira-orange (Ref. 36). The developer defined kusabira-orange as FUCCI-red, therefore we call FUCCI-red.

Comment 4;

Line 139: repetition of NIR fluorescence

Response:

Line 135: We used the word “it” as a pronoun in this sentence in the revised version.

Comment 5;

Line 185: GMB mice model should be mouse model.

Response:

Line 173: We corrected from “mice” to “mouse” in the revised version.

Comment 6;

Line 209: significantly improved disease-free survival, should mention how long the survival increased

Response:

Lines 197-199: We added how long survival increased in the revised version.

Comment 7;

Lines 225-231: should be more succinct

Response:

Lines 214-215: We made this section more succinct in the revised version.

Comment 8;

Line 239: would be helpful to shortly mention the findings of the studies cited.

Response:

Lines 218-221: We have briefly mentioned the findings of the cited studies in the revised version.

Comment 9;

Line 244: intratumorally and intravenously should be intratumoral and intravenous injection 

Response:

Line 225: We have made these corrections in the revised version.

Comment 10;

Line 251: VATS maker should be marker

Response:

Line 232-233: We deleted “VATS” in the revised version.

Comment 11;

Lines 275-288: the paragraph is about PDX but does not explain what host species has been used until the last sentence.

Response:

Lines 262 and 264: We have stated that PDX was human pancreatic cancer in nude mice in the revised version.

Comment 12;

Line 288: mice should be mouse

Response:

Line 266: We have made this correction in the revised version.

Reviewer 2 Report

Comments

This manuscript reviews the combination of imaging technology and oncolytic virus-therapeutic for administrating therapeutic agents, fluorescence-guided surgery, and an increased cancer fluorescent photosensitizer. This may represent a new direction in cancer therapy.

  1. In Abstract, the authors claimed that “imaging technology and our unique oncolytic adenoviruses provide …, and a precise cancer-specific fluorescent photosensitizer in the clinic.” It is not clearly whether these achieved with experimental animals or in clinical application. Also several figures, as well as some statments in some sections, imply this approach being used in patients. If the approach that you are studying is still in preclinical stage with the clinical potential, this should be clearly clarify to avoid misunderstanding.

  1. In Abstract, line 21 “they conditionally replicate only in cancer cells.” To be accurate, the word “only” should be deleted. Although all oncolytic adenoviruses are designed and selected for preferential replication in cancer cells, the replication cannot be 100% blocked or inhibited in normal cells. The hTERT promoter, driven in the vectors of authors’ studies, generally expressed highly in most cancers, but still expresses some levels in many normal cell.

  1. In line 110-112, “Adenovirus is known to force the cell cycle of the infected cells in the S/G2 phase to replicate viral DNA using host organelles, and subsequently to kill them with viral destruction of infected cells [40].” A question is how or why adenovirus can force cancer cells into S/G2 phase, as this is the base of the authors’ studies. There was a study showing that viral E1 genes can induce cyclin E and other cell cycle-related genes for forcing quiescent cancer cells into S-like phase (J Virol 82: 3415–3427.). This reference would be helpful to the review.
  2. From line 168, “The glioma was more effectively resected with high-168 dose OBP-401-based FGS than with low-dose OBP-401-based FGS or BLS. Although 12 of 14 mice that underwent BLS had recurrence of tumors, 5 of 14 that underwent OBP-401-based FGS had recurrence, low-dose OBP-401-based FGS reduced the rate of local recurrence (p = 0.002), and high-dose OBP-401-based FGS resulted in no recurrence.”  Why high-dose OBP-401 resulted in better outcomes than low-dose. Is it because that tumors could be completely resected with better cancer-specific bright illumination, or any tumor cells left in the surgical bed would be more likely destroyed by viral oncolysis, or both as OBP-401 has dual functions?
  1. The review paper presents the several advantages of their approach, but lacks the discussion of possible limitations, especially in potential clinical applications. In many studies human cancer cells are used to generate tumors in mouse. As human adenovirus can more efficiently infect and replicate in human cells (cancer or not) than in mouse cells, therefore can indicate a precise margin between the human tumor and mouse normal tissue. When used in patients, adenoviruses may infect both human cancer cells and normal tissue cells, this could be one of challenges.

Author Response

Reviewer #2

Summary

This manuscript reviews the combination of imaging technology and oncolytic virus-therapeutic for administrating therapeutic agents, fluorescence-guided surgery, and an increased cancer fluorescent photosensitizer. This may represent a new direction in cancer therapy.

Comment 1;

In Abstract, the authors claimed that “imaging technology and our unique oncolytic adenoviruses provide …, and a precise cancer-specific fluorescent photosensitizer in the clinic.” It is not clearly whether these achieved with experimental animals or in clinical application. Also several figures, as well as some statements in some sections, imply this approach being used in patients. If the approach that you are studying is still in preclinical stage with the clinical potential, this should be clearly clarify to avoid misunderstanding.

Response:

Lines 32-33: We have clarified that these experiments are in nude mice.

Comment 2;

In Abstract, line 21 “they conditionally replicate only in cancer cells.” To be accurate, the word “only” should be deleted. Although all oncolytic adenoviruses are designed and selected for preferential replication in cancer cells, the replication cannot be 100% blocked or inhibited in normal cells. The hTERT promoter, driven in the vectors of authors’ studies, generally expressed highly in most cancers, but still expresses some levels in many normal cells.

Response:

Line 21: We have corrected this section to state that oncolytic viruses have preferential replication in cancer cells.

Comment 3;

In line 110-112, “Adenovirus is known to force the cell cycle of the infected cells in the S/G2 phase to replicate viral DNA using host organelles, and subsequently to kill them with viral destruction of infected cells [40].” A question is how or why adenovirus can force cancer cells into S/G2 phase, as this is the base of the authors’ studies. There was a study showing that viral E1 genes can induce cyclin E and other cell cycle-related genes for forcing quiescent cancer cells into S-like phase (J Virol 82: 3415–3427.). This reference would be helpful to the review.

Response:

Lines 102-104: We suggested the mechanism of decoy of quiescent cancer cells to S-phase in the revised version.

Comment 4;

From line 168, “The glioma was more effectively resected with high-dose OBP-401-based FGS than with low-dose OBP-401-based FGS or BLS. Although 12 of 14 mice that underwent BLS had recurrence of tumors, 5 of 14 that underwent OBP-401-based FGS had recurrence, low-dose OBP-401-based FGS reduced the rate of local recurrence (p = 0.002), and high-dose OBP-401-based FGS resulted in no recurrence.”  Why high-dose OBP-401 resulted in better outcomes than low-dose. Is it because that tumors could be completely resected with better cancer-specific bright illumination, or any tumor cells left in the surgical bed would be more likely destroyed by viral oncolysis, or both as OBP-401 has dual functions?

Response:

Lines 164-168: We have stated in the revised version that OBP-401 had two effects in these experiments; tumor illumination and tumor ablation.

Comment 5;

The review paper presents the several advantages of their approach, but lacks the discussion of possible limitations, especially in potential clinical applications. In many studies human cancer cells are used to generate tumors in mouse. As human adenovirus can more efficiently infect and replicate in human cells (cancer or not) than in mouse cells, therefore can indicate a precise margin between the human tumor and mouse normal tissue. When used in patients, adenoviruses may infect both human cancer cells and normal tissue cells, this could be one of challenges.

Response:

We have stated in the revised version the limitation that the adenoviruses may replicate in normal tissues in human.

Round 2

Reviewer 1 Report

The manuscript has been improved form the previous version. However, there are still numerous errors that should be revised and corrected. I have made these suggestions in the PDF of the manuscript as it will be easier to follow. The parts of the text are highlighted in purple and the comments are on the side in purple text. I would like to mention that these are still not expensive corrections and the manuscript should be more carefully reviewed for English language and style. 

Author Response

Yano et al.: “Real-time fluorescence image guided oncolytic virotherapy for cancer treatment (MS No. IJMS 1028845)

Response;

The authors are grateful for the very helpful comments by the reviewer that enabled us to significantly improve the manuscript.

Reviewer #1

Summary

The manuscript has been improved form the previous version. However, there are still numerous errors that should be revised and corrected. I have made these suggestions in the PDF of the manuscript as it will be easier to follow. The parts of the text are highlighted in purple and the comments are on the side in purple text. I would like to mention that these are still not expensive corrections and the manuscript should be more carefully reviewed for English language and style. 

Response:

We corrected errors as Reviewer #1 indicated. Moreover, the revised manuscript was carefully reviewed and corrected by a native English scientist (co-author).
